# Sex-Based Mechanisms of Cardiac Development and Function: Applications for Induced-Pluripotent Stem Cell Derived-Cardiomyocytes

**DOI:** 10.3390/ijms25115964

**Published:** 2024-05-29

**Authors:** Yinhan Luo, Sina Safabakhsh, Alessia Palumbo, Céline Fiset, Carol Shen, Jeremy Parker, Leonard J. Foster, Zachary Laksman

**Affiliations:** 1Centre for Heart Lung Innovation, Department of Medicine, University of British Columbia, Vancouver, BC V6Z 1Y6, Canada; hattie02@student.ubc.ca (Y.L.); jeremy.parker@ubc.ca (J.P.); 2Centre for Cardiovascular Innovation, Division of Cardiology, University of British Columbia, Vancouver, BC V6T 2A1, Canada; sinasafa@student.ubc.ca; 3Michael Smith Laboratories, Department of Biochemistry & Molecular Biology, University of British Columbia, Vancouver, BC V6T 1Z4, Canada; apalumbo507@gmail.com (A.P.); foster@msl.ubc.ca (L.J.F.); 4Research Centre, Montreal Heart Institute, Faculty of Pharmacy, Université de Montréal, Montréal, QC H1T 1C8, Canada; celine.fiset@umontreal.ca; 5Department of Integrated Sciences, University of British Columbia, Vancouver, BC V6T 1Z2, Canada; carol530@student.ubc.ca

**Keywords:** human-induced pluripotent stem cell, cardiomyocyte, sex hormone, sex hormone receptor, cardiac development, cardiology

## Abstract

Males and females exhibit intrinsic differences in the structure and function of the heart, while the prevalence and severity of cardiovascular disease vary in the two sexes. However, the mechanisms of this sex-based dimorphism are yet to be elucidated. Sex chromosomes and sex hormones are the main contributors to sex-based differences in cardiac physiology and pathophysiology. In recent years, the advances in induced pluripotent stem cell-derived cardiac models and multi-omic approaches have enabled a more comprehensive understanding of the sex-specific differences in the human heart. Here, we provide an overview of the roles of these two factors throughout cardiac development and explore the sex hormone signaling pathways involved. We will also discuss how the employment of stem cell-based cardiac models and single-cell RNA sequencing help us further investigate sex differences in healthy and diseased hearts.

## 1. Introduction

Important differences have been identified when comparing the structure and function of male and female hearts. The contributions of both sex chromosomes and sex hormones in this dimorphism have been studied in depth [1]. We have also come to appreciate that these and other factors in sex-specific cardiac dimorphism vary throughout developmental stages. During embryogenesis, the earliest signals that lead to sex-specific differences in cardiac development originate from genomic chromosomal differences between the sexes [2]. As development proceeds and cardiac structures gain more complexity, sex steroid hormones start having an increasingly dominant effect on the sexually dimorphic properties that govern cardiac structure and function [3,4]. The mechanisms by which sex hormones act continue to change throughout development, impacting multiple layers of cardiac physiology. This review will discuss the evolution of male and female differences in cardiac structure and function with an emphasis on embryogenesis and early development.

It must also be noted that the actions of sex hormones, which result in a broad range of structural, contractile, and electrophysiological differences between male and female cardiomyocytes, have an impact on both normal cardiac physiology and in disease conditions [5,6]. The impact of sex hormones, however, tends to gradually decline with age or with the onset of concomitant diseases that have a greater impact on structure and function, or that may in themselves alter hormonal profiles [7].

Our efforts to understand the differences between male and female cardiac structure and function have employed several model organisms. Some of the most recent investigations have focused on engineered stem cell technology to develop human and patient-specific cardiomyocytes in vitro [8]. This field has rapidly increased in complexity, not only in terms of relevant physiologic or molecular assays, but also in the study of more complex systems that may include multiple cell types, multiple cell lines, and tissue engineering, all to recreate the most accurate human model as a disease in a dish [9]. These and other innovative approaches will be discussed as potential future directions towards a better understanding of sex-specific mechanisms of cardiac physiology in health and disease.

## 2. Cardiac Development and Sex-Specific Differences

As the first organ to form and function in embryogenesis, the development of the human heart begins in gestational week 2, and the four-chambered heart is formed by week 8 to 9 to supply oxygen and nutrients to the rest of the embryo [10,11].The anatomical structures and genetic markers driving human heart morphogenesis are well studied with the use of techniques such as magnetic resonance imaging, immunofluorescence, and in-situ hybridization [12,13]. Progenitors including myocardial and proepicardial cells from the mesoderm and cardiac neural crest cells from the ectoderm undergo specification and differentiation while migrating to form the cardiac crescent and fuse to become the primitive cardiac tube (Figure 1) [10,11,14]. The first beat of this linear heart tube is generated at around day 22 [10,14]. Upon the rightward looping of the cardiac tube in week 4, the elongated outflow tract and the ventricles start to emerge while the atria also begin to differentiate [11]. Finally, the four-chambered heart is formed after atrial and ventricular septation during gestational weeks 6 to 9 [12]. At this stage, the atrioventricular valves and the semilunar valves are formed to maintain the unidirectional blood flow through the heart [12]. More recently, the size, volume, and growth patterns of the developing human heart have also been quantified using immunohistochemistry and 3D reconstruction, demonstrating its exponential growth throughout embryogenesis [15].

The developmental origin of sex-based differences in the heart occurs before gonad formation, with differentially expressed transcriptional and epigenetic factors throughout cardiogenesis [16,17]. A study conducted on mouse models have shown that sex chromosomes are responsible for the differences between male and female hearts at the earliest stage of cardiac development. Turner syndrome (XO) and Klinefelter mouse models (XXY) demonstrated that 5% of the differentially expressed cardiac proteins between males and females are located on the X chromosome [2]. These include DEAD-box helicase 3, a transcriptional and translational regulator, and ubiquilin 2, which is involved in protein degradation. These proteins act in a dose-dependent manner, while none are found on the Y chromosome [2]. A subset of the differentially expressed proteins are conserved in adulthood, such as the ones involved in cardiac conduction and CM contraction in males as well as those associated with metabolic processes in females [2]. Even though many sex-specific markers are located on autosomes, the mechanisms of sex-specific autosomal gene expression remain unknown [17]. In addition, X chromosome inactivation (XCI), which is initiated in the two to eight-cell stage of embryonic development, may contribute to sex differences in cardiac phenotypes, but this needs further investigation as the X chromosome traditionally has not been included in genome-wide studies [2,18]. It is now known that differentially expressed transcripts from the X chromosome can be involved in DNA or histone methylation. Epigenetic regulation caused by differences in the expression of methylases and demethylases is crucial to heart development [19]. These genes are usually X escape genes, which are genes on the X chromosome that are not impacted by XCI and are more highly expressed in the XX complement than the XY complement [20]. Since there are established connections between DNA methylation and sex differences in blood lipid levels, stroke risk, and cardiometabolic pathways, sex-based differences in the activity of these pathways likely affect cardiovascular health and function on a cellular level [21].

Moreover, sex chromosomes influence heart function beyond the production of gonads and gonadal sex hormones [22]. A recent study using mouse models has determined that sex chromosome complement pathways establish sex differences in cardiovascular function prior to gonad development [2]. The differential expression of proteins prior to gonad development and sex hormone production in a fetus affect the development of cardiovascular tissues and structures. A notable protein that is enriched in both embryonic and adult female cardiomyocytes (CMs) is the Alpha-1-B glycoprotein (A1BG) [2]. This glycoprotein is enriched in embryonic and adult CMs and the loss of A1BG results in cardiac abnormalities in XX but not XY mammals [2]. The effect of sex chromosome complement had been tested in numerous studies using four core genotype (FCG) mice [23]. These mice have the *Sry* gene deleted from the Y chromosome and inserted into an autosome, so that the mice develop testes regardless of XX or XY complementation. These studies report that while many proteins segregate with the development of ovaries or testes, over 100 proteins were differentially expressed solely based on the sex chromosome complementation. The XX genotype with testes has similar protein expression as the XX genotype with ovaries, while the same pattern applies to the XY genotype, independently of the gonadal phenotype [24]. While sex hormone production has a significant impact on cardiovascular development and function, the effects of sex chromosome complementation exist beyond gonad development and the production of sex hormones in the gonads [2,22,24].

Most studies on cardiac sex disparities have focused on the roles of sex hormones. These are produced and begin to act on the developing heart after gonad formation [2,5,7]. Cells in the differentiating testis start to secrete androgens by week 7 of gestation, whereas the fetal ovary can convert androgen to estrogen by week 12 [3,4]. Subsequently, released sex hormones are taken up by receptors on the fetal heart to drive sex differences [3,4]. A study in rats has shown that the female fetal heart has a significantly higher abundance of estrogen receptor alpha (ERα) and beta (ERβ) than the male counterpart [25]. Interestingly, testosterone treatment in fetal sheep hearts increased the expression of ERα and ERβ but not the androgen receptor (AR) in both male and female sheep [26]. The expression of the androgen receptor in prenatal mouse CMs has also been demonstrated, suggesting the involvement of androgens during cardiogenesis [27]. Despite evidence for the presence of sex hormones and their receptors during development, there is a lack of understanding on the mechanisms of action. For instance, it remains unclear how sex chromosomes drive differences in sex hormone receptor expression during development or how differences in sex hormone or receptor profiles lead to sex-based differences in cardiac function.

During puberty, male and female hearts begin to develop differently and exhibit sex-specific differences under physiological and pathophysiological conditions in adulthood [21]. The male heart is larger in size with a thicker left ventricular wall and has a larger cardiac output than the female heart [21]. However, the female heart shows increased contractility and ejection fraction (EF) and has a higher fraction of CMs than the male counterpart [21,28]. Although teenage males and females do not exhibit differences in electrocardiogram (ECG) patterns, adult males have a shorter corrected QT interval (QTc) than females, and this difference diminishes with age [28,29]. Also, males and females have variable prevalence and outcomes of cardiovascular disease (CVD). For instance, males are more likely to have severe outcomes such as sudden death due to dilated cardiomyopathy (DCM) and hypertrophic cardiomyopathy (HCM) [18]. They are also at a greater risk for developing atrial fibrillation [30]. Females on the other hand, have a higher risk for atrial fibrillation reoccurrence after ablation and torsades de pointes related to QT prolongation [18,31]. Interestingly, pre-menopausal women are less likely to have coronary artery disease than men, but the incidence rises and exceeds that in men after menopause, implicating a potentially protective role of the female sex hormone estrogen [32].

After having completed cardiac embryogenesis and development, sex hormones continue to exert actions that maintain sex-based differences in cardiac structure and function between males and females. The pathways mediating these differences throughout adulthood will be discussed below.

## 3. Sex Hormone Signaling Pathways and Mechanisms of Action

The male hormone testosterone and the female hormone estrogen are involved in numerous signaling pathways and act on cardiac cells via both genomic and non-genomic mechanisms (Figure 2) [33,34]. In the genomic pathway, the hormones bind to androgen receptors (ARs) and estrogen receptors (ERs) in the cytoplasm and migrate to the nucleus to regulate gene expression [34,35,36]. Alternatively, they can initiate a rapid non-genomic effect via G protein-coupled receptors (GPCRs) on the cell membrane [34,35,36].

### 3.1. Testosterone

One of the key effects of testosterone is cardioprotection. Testosterone elicits its cardioprotective effects through multiple pathways, many of which converge to inhibit cellular apoptosis [36,37]. In rat CMs undergoing superoxide damage, testosterone supplements of 5–100 nM increased cell survival from 50% to 70% in a dose-dependent manner [37]. This effect was shown to be AR-mediated, as pre-treatment with flutamide (a competitive AR antagonist) reduced the protective effects of testosterone. Testosterone was found to significantly increase Akt activation in CMs with normal nuclear factor kappa B (NF-κB) expression, while attenuating the overexpression of caspase 3 involved in apoptosis to promote cell survival [37]. The ERK signaling pathway was also activated by testosterone, which could lead to NF-κB activation and the transcription of antioxidant genes. Further, testosterone was found to confer cardioprotection in both isolated perfused hearts and CMs by enhancing the anti-apoptotic effect of α_1_-adrenoreceptors [38]. Another example of the AR-dependent action of testosterone is the regulation of ischemia-induced neovascularization [39]. In wildtype mice that experienced ischemia, blood flow was enhanced and hypoxia-inducible factor 1-α (HIF1A) was augmented by dihydrotestosterone (DHT), whereas these effects were abrogated or attenuated in AR knockout mice [39].

Testosterone can also induce deleterious effects and promote cardiac injury through the promotion of hypertrophy [40,41,42,43]. Cardiac hypertrophy is generally an adaptive response to pressure overload. However, it can also be maladaptive in response to exposures such as prolonged exposure to excess testosterone, which in turn leads to an increased risk of arrhythmias and heart failure [40]. Testosterone-induced hypertrophy is dependent on the crosstalk between several signaling pathways, including the activation of rapamycin complex 1 (mTORC1) via IP_3_/Ca^2+^ and MEK/ERK1/2 [41,42]. Testosterone induces CM hypertrophy by activating a critical transcription factor, NFAT, through the inhibition of GSK-3β via genomic actions [43]. Moreover, 2 h of 100 nM testosterone treatment resulted in increased reactive oxygen species (ROS) generation as well as procaspase-3 activation in rat vascular smooth muscle cells [44]. The study confirmed the proapoptotic effect of testosterone and provided evidence of testosterone-induced apoptosis via the extrinsic pathway. Another adverse effect of testosterone is enhanced cardiac remodeling after myocardial infarction (MI), leading to reduced cardiac function and an increased probability of septal rupture [45,46,47].

### 3.2. Estrogen

The two established estrogen receptors ERα and ERβ are responsible for the genomic and non-genomic actions of estrogen [48,49,50]. One group claimed that only ERβ mRNA was undetectable in mouse ventricular CMs, whereas others have reported the presence of both ERα and ERβ at the protein level in the atria and ventricles of mice and guinea pigs [51,52,53]. The receptors also vary in tissue distribution and are involved in different signaling pathways in the cardiovascular system [48,49]. Genomic actions are dependent on nuclear ERα, whereas non-genomic actions are mediated by ERα and/or ERβ localized to the plasma membrane [48,50]. Another membrane-bound G-protein coupled estrogen receptor (GPER) known as GPR30 induces acute signaling [54].

Studies into the role of circulating estrogen demonstrate mainly cardioprotective properties [55]. Membrane-bound ERα and ERβ protect the heart from injury via downstream effectors including PI3K, Akt, and eNOS [48,56]. Unlike testosterone, 17β-estradiol (E2) bound to ER inhibits NF-κB and activates PI3K/Akt to inhibit CM apoptosis [55,57]. Meanwhile, ER expression is also regulated by NF-κB as it suppresses the transcription of ERα in the heart by binding responsible promoters [56]. Estrogen can also stimulate the production of nitric oxide (NO) via the PI3K Akt pathway in endothelial cells [48]. Akt then activates mitochondrial GSK-3β [55], which elicits the opposite effect of testosterone. A third ER, known as GPR30, is a GPER which contributes to the rapid response of estrogen by activating the same pathways via a truncated ERα which is mainly present in the plasma membrane [48,49,55,57,58]. Under ischemia or reperfusion injury, GPR30 is an essential modulator of E2 for acute cardioprotection [59]. The acute protective effects of E2 were lost in GPR30 knockout, while knocking out other ERs did not make a difference in the infarct size or functional test results [59]. GPR30 is also required for the increase in the mitochondrial Ca^2+^ retention capacity mediated by E2 [59,60]. The findings also support the activation of Akt and ERK1/2 and the deactivation of GSK-3β as a result of E2/GPR30-induced cardioprotection [59]. Furthermore, E2 regulates the expression of protective heat shock proteins [48,55]. An example is HSP72, which stabilizes the mitochondrial membrane to help reduce apoptosis [55]. In a mouse model of heart failure, E2 restores the EF of mouse hearts with pre-existing heart failure through Erβ [61]. EF significantly improved after ERβ agonist treatment, while ERα agonist treatment did not make a difference. Another study showed that supplementing E2 counteracted the prolongation of action potential duration (APD) in a guinea pig heart failure model [62]. Also, the anti-hypertrophic action of E2 was found to be mediated by ERβ as the ERβ agonist treatment led to decreased cardiac hypertrophy, measured by heart weight to body weight ratio [61]. Furthermore, cardiac fibrosis in heart failure was attenuated by E2 via Erβ, as demonstrated by the decreased expression of profibrotic markers [55].

## 4. The Effects of Hormonal Variation on Cardiac Status

Hormonal variations can occur due to physiological processes such as menopause or general aging. They can also occur due to underlying pathology that may or may not be sex-specific. Due to the systemic nature of circulating hormones, these variations affect multiple organs including the heart, leading to changes in cardiac structure and function [7].

It has been repeatedly demonstrated that endogenous testosterone levels in males decline physiologically with age as testicular Leydig cells progressively lose turnover capacity [63,64]. The decline in endogenous testosterone can also arise in pathological states such as hypogonadism [65,66]. The effects of declining testosterone levels on cardiac physiology have been studied via orchiectomy (ORC) experiments. These experiments involve the bilateral removal of male testes and rearing under testosterone-deficient or testosterone-repleted conditions [67]. The echocardiography of ORC male mice grown to 10 weeks demonstrated reduced systolic function and concentric left ventricular remodeling compared to testosterone-repleted mice [68]. In other experiments involving ORC and sham operations on male mice, the assessment of electrophysiological dynamics showed prolonged calcium transients and reduced calcium transient amplitudes in ORC vs. sham-operated mice [69,70]. Subsequent Western blot analyses demonstrated significantly higher levels of phospholamban (PLN) in the ORC mice. PLN is a regulatory protein which acts to modulate the activity of sarco(endo)plasmic reticulum Ca^2+^-ATPase (SERCA) [71]. SERCA is a transmembrane calcium transporter located in the sarcoplasmic reticulum that is crucial for cardiac calcium handling and is a key component of cardiac excitation–contraction coupling (ECC) [72]. Therefore, increased levels of PLN regulate SERCA activity levels to the point of reducing calcium transient amplitudes and prolonging calcium transients during periods of chronic testosterone deficiency. The precise mechanism through which this occurs remains to be fully elucidated. Lastly, further electrophysiology experiments on ventricular myocytes isolated from male mice at 22–28 months, who had undergone either ORC or sham surgery at 1 month, revealed a prolonged APD in ORC CMs associated with an increased late inward sodium current (I_Na,L_) [73]. This prolongation of the APD was associated with more triggered activity (early/delayed afterdepolarizations) and a higher rate of arrhythmic events. The precise mechanism through which declining testosterone levels contribute to increased inward sodium currents that prolong the APD and increase the risk of arrhythmias is not yet clear. However, these studies suggest that variations in testosterone levels have a notable impact on cardiac structure and function that must be considered with normal physiologic changes in its levels such as with aging [74,75].

There are also age-dependent variations in the major female sex hormone estrogen [76]. The ovaries, which are the key producers of estrogen in women, may reach a natural senescence, termed menopause, which typically occurs between the fourth and fifth decades of life. Concordant with declining follicular activity, estrogen levels decline in the years preceding menopause and reach a plateau once a lower limit threshold of follicular activity is reached. Similar to studies in male animal models, ovariectomies (OVX) in females allow for an understanding of the molecular changes that occur with declining estrogen levels [58,67]. Interestingly, echocardiographic experiments with female mice of 24 months who had either undergone OVX or sham operations at 1 month showed no significant changes in cardiac structure other than a reduced interventricular septal end diastole (IVSd) in OVX mice when compared to sham-operated ones [77]. However, calcium imaging of CMs isolated from these two groups demonstrated increased sarcoplasmic reticular calcium stores and a higher frequency of spontaneous calcium release in OVX-treated females [77]. Dysregulations in intracellular calcium handling secondary to estrogen deficiency in females post-OVX were confirmed in several follow-up studies [78,79]. Subsequent experiments involving high-resolution calcium imaging and protein quantification demonstrated no difference in the expression levels of key cardiac calcium handling proteins such as ryanodine receptor (RyR) and sodium-calcium exchanger (NCX) in the hearts of female mice post-OVX vs. sham operations [75,80]. However, calcium flux through these channels was significantly different between these two groups [80]. This functional difference in calcium channel activity was shown to be related to increased activity levels of a key kinase called protein kinase A (PKA), which enhanced the activity of downstream cardiac ion channels through the post-translational phosphorylation of allosteric sites. Interestingly, it has also been shown that some of these changes after OVX in females are reversible with estrogen replacement [79]. However, these findings have yet to be confirmed with follow-up experiments. Overall, these data demonstrate that declining estrogen levels mainly exert a functional effect on cardiac physiology by altering intracellular calcium handling.

Furthermore, there is now evidence to suggest that sex hormone signaling alongside genetic and epigenetic processes will ultimately result in sex-based differences in mature human CMs that manifest as unique electrophysiological and contractile properties [6]. A basic understanding of these sex-based differences will be crucial in improving our assessment of how they can be uniquely affected in patients with CVD, regardless of sex.

## 5. Sex-Based Differences in Mature Human Cardiomyocytes

Differential sex hormone signaling over time and across biological sexes is one mechanism through which physiological differences in human CMs arise [5,81]. However, genetic and epigenetic mechanisms such as sex chromosome dosage effects and differences in X or Y-chromosome inactivation lead to variations in gene expression patterns across organ systems including the heart [82,83,84,85]. An analysis of the pathways affected by differential gene expression shows unique differences in cardiac physiology between males and females. Key affected mechanisms include cardiomyocyte contractility, calcium handling, and electrophysiology.

Experiments with mature human CMs have revealed increased cardiac contractility in CMs from females compared to those from males [86]. Further investigation into the underlying molecular mechanisms showed that these findings are related to the increased expression of contractile proteins and a faster activation rate of relevant regulatory kinases [5,86,87]. For example, mRNA levels of key contractile proteins such as α- and β-myosin heavy chain (MHC) proteins have been shown to be significantly higher in cardiomyocytes isolated from females relative to age-matched males [86]. Moreover, bulk RNA sequencing of CMs from females vs. males has also revealed the increased expression and activity of several kinases involved in mediating this observed difference in contractility between male and female CMs [5]. For example, PKA is one kinase that was found to be significantly upregulated in CMs from females [5]. Subsequent functional studies demonstrated the strong involvement of PKA in regulating activity levels of contractile proteins such as α- and β-MHC via phosphorylation [88]. However, the precise mechanism through which differential sex hormone exposure or sex-specific gene expression leads to the increased expression of contractile proteins or heightened activity of relevant kinases is yet to be fully understood [5]. The observed difference in cardiomyocyte contractility between males and females has been validated clinically through echocardiography and cardiac magnetic resonance data that show an increased EF in females compared to males [89,90]. Furthermore, it has been postulated that this sexual dimorphism in cardiac contractility may contribute to the disproportionately high rates of heart failure with preserved EF and worse associated outcomes in females [91].

Calcium handling is another process found to differ between female and male CMs [6,92]. Several papers have reported differences in expression levels and functional states of relevant cardiac proteins including ryanodine receptor 2 (RyR2), SERCA, and NCX [80,93]. Much of these data come from studies of mouse CMs which have implicated the differential expression of several proteins to account for differences in calcium handling [52,75,94,95]. However, one protein with a critical role in cardiomyocyte calcium handling found at significantly different expression levels in females versus males is the voltage-dependent L-type calcium channel 1 (Ca_v_1) [96]. This is a five-subunit channel complex that resides at the cardiomyocyte surface membrane and passes inward Ca^2+^ current (I_Ca,L_) upon depolarization to allow for ECC. The alpha 1C subunit (Ca_v_1.2) is the primary subunit in the heart and its expression levels are used as surrogates of overall Ca_v_1 density [97]. Using post-mortem human left ventricular tissue and via antibody-based staining for Ca_v_1.2, it has been shown that the expression levels of this key subunit are higher in females relative to males in a region-dependent manner [96]. Follow-up experiments have revealed that this occurs through estrogen binding and the activation of membrane-bound estrogen receptors and initiation of a signaling cascade involving the PI3K-Akt pathway [98]. This leads to phosphorylation of the transcription factor cAMP Response Element-Binding Protein (CREB), which then binds promoters upstream of the gene encoding Ca_v_1.2, CACNA1C, and upregulates its expression. Overall, the sex-based differences in cardiomyocyte calcium handling contributes to several downstream processes including ECC [99]. This is likely one of several reasons for the observed differences in the rate and presentation of CVD in females versus males [100].

Lastly, mature human CMs also have sex-specific differences in cardiac electrophysiology (Figure 3) [101]. Efforts to identify these findings originated from consistent clinical observations of prolonged corrected QT (QTc) intervals in females versus males [102]. Following these clinical observations were basic experiments aimed at characterizing the underlying molecular mechanisms involved. These experiments were performed on ventricular CMs from mice and demonstrated a longer APD in ventricular CMs in females relative to males [94]. Based on the significant contribution of the phase 2 repolarization phase of the cardiac action potential, subsequent experiments were performed to screen for changes in the expression or function of ion channels involved in phase 2 repolarization. These studies showed reduced ultrarapid delayed rectifier potassium currents (I_Kur_) in females relative to males [103]. This was mediated by relatively increased expression and activity levels of the potassium channel Kv1.5 in males which was mediated by androgen exposure over time. These findings were confirmed via mouse studies involving castrated and control male mice [103,104]. Furthermore, subsequent experiments in ovariectomized and control female mice alongside control male mice showed that estrogen exposure to ovariectomized female mice resulted in the reduced expression of Kv1.5 transcripts and downstream channel activity [105]. The precise underlying genomic mechanisms through which the activity of estrogen or the reduced activity of androgens leads to the reduced expression of the Kv1.5 potassium channel, lowers I_Kur_, and prolongs the APD to ultimately yield a prolonged QTc in females, are not fully understood. Importantly, the presence of conflicting data showing increased potassium channel activity in OVX female mice suggest that sex-specific differences in QTc intervals may be driven by differences in the activity of other key channels or proteins that are not yet identified [106].

In the mature human heart, both sex chromosome complementation and sex hormones contribute to the regulation of protein expression by DNA methylation. The bioavailability of both testosterone and estradiol are correlated with DNA methylation [107,108]. Epigenome-wide association studies using DNA extracted from peripheral blood samples have shown that testosterone introduction causes changes in DNA methylation, and both ERs are known regulators of passive and active DNA regulation [107,108]. Furthermore, a recent study on transgender people undergoing gender-affirming hormone therapy (GAHT) determined that DNA methylation was differentially regulated prior to starting hormone therapy and at the 6- and 12-month marks for both masculinizing and feminizing hormone therapy. The resulting DNA methylation signatures for transmasculine and transfeminine people undergoing GAHT were unique from the signatures expected in cisgender males and females, indicating that hormones are not the only influence on this regulation [109]. From this, it can be concluded that mature human cardiovascular function is influenced by the interplay of both sex hormones and chromosomes and that more research is needed to determine what is influenced by these factors. The understanding of regulation through DNA methylation in cardiovascular tissues could provide further insight into sex differences in gene regulation and metabolic pathways that influence heart health and disease susceptibility.

## 6. Using Human-Induced Pluripotent Stem Cell-Derived Cardiomyocytes (hiPSC-CMs) to Better Understand Sex-Based Differences in Cardiac Function and Dysfunction

Animal models have been used extensively in cardiovascular research and have provided insights into the physiology and pathophysiology of various cardiac conditions. Nonetheless, the cardiovascular system of each species evolved differently to meet their specific needs and animal hearts do not closely resemble the anatomy or physiology of the human heart. For example, the beating rate, contractility, and oxygen consumption of rodent hearts all differ from human hearts, and the types and expression levels of ion channels vary across species. The use of large animals also raises significant ethical concerns, in addition to being quite costly [110]. Human samples, on the other hand, are ideal but their availability is very limited and CMs do not proliferate or survive in long term cultures [111]. Fortunately, recent advances in hiPSC-CMs are revolutionizing the field of cardiac research [8]. These hiPSC-CMs present a promising platform for in vitro disease modeling and drug screening as they are patient-specific and capable of generating reliable and reproducible results (Figure 4) [8,112,113]. Meanwhile, standardized hiPSC-CM ventricular and atrial differentiation and expansion protocols have made the cell model robust and scalable [114,115,116,117].

The advantages of hiPSC-CMs have led to their growing popularity in cardiac disease modeling. These hiPSC-CM models have been employed in the study of a wide variety of cardiomyopathies. In brief, models established for inherited arrhythmias such as long QT syndrome are used to study the action potential abnormalities and the changes in ionic currents in disease states [30,31]. The phenotypes of structural cardiomyopathies including HCM, DCM, and arrhythmogenic cardiomyopathy (ACM) have all been demonstrated using hiPSC-CMs from patients harboring disease-causing genetic variants [118,119]. Acquired diseases such as heart failure have also been investigated using hiPSC-CM models [118,120]. Ischemic cardiomyopathies can be modeled by culturing the hiPSC-CMs in an ischemia–mimetic solution in hypoxic conditions [119]. Even less common metabolic cardiomyopathies such as Pompe disease and Barth syndrome have also been successfully modeled using hiPSC-CMs [118]. Known for its accurate proarrhythmia risk prediction, and the electrical and structural recapitulation of the disease phenotype, the use of these models in studying cardiac diseases continues to expand [113,118,119,121]. With increased knowledge of the intrinsic differences in cardiology between male and female and sex-biased disease progression, efforts have been made to improve the hiPSC-CM model by making it sex-specific. In 2019, a qualitative study showed that male and female hiPSC-CMs exhibited different electrophysiological responses to various pharmaceutical drugs. For example, the QT-prolonging drug dofetilide consistently triggered early afterdepolarizations (EADs) in female hiPSC-CMs but no EADs were observed in male cell lines when exposed to the same dose of drug [122]. The genetics and biological sex of the iPSC donors as well as the effects of sex hormones on cardiac diseases have been investigated [22,123,124,125].

In arrhythmia research, hiPSC-CMs were employed in the study of two ion channels in LQT syndrome type 2 where female and male-derived cells were treated with E2 [96]. It was found that the female hiPSC-CMs responded to estrogen with an upregulation in I_Ca,L_ and sodium–calcium exchange (I_NCX_) currents, yet male samples did not. Similarly, hiPSC-CMs were utilized in a study on the sex differences in drug-induced LQT and susceptibility to proarrhythmias [124]. For instance, dofetilide-induced APD prolongation was significantly greater in female-derived than male-derived hiPSC-CMs, demonstrating the higher sensitivity of the female heart to delayed rectifier potassium current (I_Kr_) blocker-induced APD prolongation. The addition of 10 nM E2 led to an increase in field potential duration (FPD) in both male and female hiPSC-CMs and a decrease in FPD was observed after DHT treatment (40 nM). These results indicated that sex hormones can cause the expected physiological effects in hiPSC-CMs but not in a sex-specific manner [124]. In addition, nodal-like hiPSC-CMs (N-iPSC-CMs) have been differentiated to study the effects of pregnancy and estrogen on cardiac automaticity. E2 treated N-hiPSC-CMs exhibited significantly higher hyperpolarization-activated current (I_f_) density, increased diastolic depolarization, and accelerated action potential firing rate, aligning with the findings from the mouse model [126]. The electrophysiological effects of DHT were also tested in a study on androgen deprivation therapy-related acquired LQT [125]. A total of 30 nM of DHT shortened the APD at 90% repolarization (APD_90)_ of the hiPSC-CMs after an acute exposure of 15 min), and DHT was also able to reverse the APD prolongation caused by enzalutamide, an androgen receptor inhibitor [125].

In addition to their electrophysiological effects, sex hormones also play an essential role in the pathogenesis of cardiac diseases. In an hiPSC-CM-based arrhythmogenic right ventricular (ARVC) model, pre-menopausal estradiol levels slowed down the lipogenesis and apoptosis of CMs, whereas high levels of testosterone accelerated the disease progression, supporting clinical findings [127]. The cardioprotective role of estrogen has been evaluated in stress-induced cardiomyopathy, known as Takotsubo syndrome, which predominantly affects postmenopausal women. It was discovered that estrogen-activated GPER led to the alleviation of epinephrine-induced cardiac injury [128]. Furthermore, a study of Takotsubo syndrome-associated LQT revealed that estradiol helps protect hiPSC-CMs against the toxic effects of catecholamines [129]. Estradiol-treated cells also prevented isoprenaline-induced APD prolongation and suppressed ROS production [129].

Most sex-specific hiPSC-CMs studies focused on the effects of sex hormones on the electrophysiology of the cells. The changes in action potential and currents are measured either at the single cell level using patch clamp techniques [96,125,128,129] or at the monolayer level with CardioECR (extracellular recording) or Maestro MEA (multi-electrode arrays) techniques [124]. However, the mechanism of action of sex hormones is lacking in these studies. Whether the hormones act in the genomic or non-genomic pathway and how they contribute to the disease progression remains widely unknown. In fact, only one study assessed the changes in ERα and AR expressions in hiPSC-CMs using qPCR [124]. Despite the potential of hiPSC-CMs in sex-specific cardiac disease modeling, the current studies still emphasize the findings in animal studies and use hiPSC-CMs as a support or validation. To become a more reliable in vitro model with broader applications, the hiPSC-CM model needs to overcome some limitations, and the future lies in the advances in sequencing technologies and multi-omics approaches.

## 7. Future Directions in Understanding Sex-Specific Cardiac Diseases

The limitations of currently available technologies and unanswered questions about sex-specific cardiac dysfunction will guide the future work in the field. While hiPSC-CM models have been instrumental in advancing our understanding of the sex-specific mechanisms of cardiac physiology, their shortcomings present areas for future improvement. First, hiPSC-CM models are typically only comprised of CMs without other accompanying cell types in the natural environment of the heart such as fibroblasts [130]. This prevents the capture and assessment of the extensive crosstalk that occurs between different cell types in cardiac tissues in vivo [131]. Current approaches to address this issue have focused on co-culturing hiPSC-CMs with an increasing number of different cell types found in cardiac tissues in vivo in an effort to better model this organ system in health and disease [9]. Moreover, hiPSC-CMs display contractile and electrophysiological properties that are fetal-like and less akin to the characteristics of mature human CMs [132]. This limits the translatability of findings from hiPSC-CMs to adult human patients with cardiac disease, regardless of the sex. Progress in this area is being made by targeting different transcriptional and signaling factors or co-culturing of hiPSC-CMs with different agents to hasten cardiomyocyte maturation [133,134,135].

Apart from improvement of hiPSC-CM models to study sex-specific cardiac disease, other directions of advancement in this area include the studying of cellular transcriptional changes in response to differential sex hormone exposure between the sexes [136]. Currently, the most advanced tool available for this purpose is single-cell RNA sequencing (scRNAseq). By providing transcriptomic information on a single-cell level, this approach offers an unprecedented degree of understanding about the effects of prolonged sex-hormone exposure on individual cells and cell–cell networks involving different cell types [137,138]. For example, Skelly et al. used scRNAseq to identify sexually dimorphic genes with divergent directionality in different cell populations in the mouse heart [139]. The differential gene expression amongst different cell types between females and males establishes a basic framework for understanding differences in the presentation of CVD between males versus females. Moreover, McLellan et al. used scRNAseq to identify the transcriptomic response by the heart to chronic stress which was induced via a 2-week continuous administration of the profibrotic stimulus angiotensin II [140]. They found a sexually dimorphic transcriptional response to chronic angiotensin exposure, identifying the basic mechanisms underlying the sex-specific characteristics observed clinically in patients with prolonged hypertension, which is a well-known trigger for angiotensin release in humans [141]. These studies demonstrate the power of scRNAseq in understanding sex-specific differences in CVD and provide the basis for follow-up experiments in this area. For example, as one of the most common cardiac disorders, the mechanisms by which atherosclerotic coronary disease initiates and progresses in females versus males have yet to be identified. Also, a scRNAseq assessment of electrophysiological differences between males and females has also not yet been performed. This would provide the foundations for understanding sex-specific mechanisms of arrhythmia onset and progression, potentially providing new avenues for targeted therapies.

Lastly, other approaches to characterize the sex-specific dimorphism in CVD onset and progression are multi-omic studies that integrate several layers of molecular biology such as genomics, transcriptomics, and proteomics [142]. By incorporating multiple underlying mechanisms, these approaches enable a more comprehensive understanding behind sexually dimorphic CVD onset and progression. These methods have been used to study cardiomyopathies and other conditions in cardiology in general. For example, through a case–control study of patients with DCM at the transcriptional, genetic, and epigenetic levels, Meder et al. identified novel epigenetic loci in DCM pathogenesis which carry the potential for clinical translation as biomarkers for disease progression [143]. Similar work has not yet been reported for studying sex-specific mechanisms of CVD but do provide insights into the type of experiments that are currently under way [144]. For example, a multi-omic analysis of cardiac tissue from males and females with coronary artery disease and similar background clinical profiles may reveal sex-specific mechanisms for the development of biomarkers that can be used diagnostically or prognostically or for developing targeted interventions.

## 8. Conclusions

Distinctions between the sexes with regards to the presentation and outcomes of CVD have become increasingly apparent. Research into the underlying mechanisms behind these observations have revealed a wide range of sexually dimorphic processes that include cardiac electrophysiology and calcium handling. Despite these advancements, there remain areas for investigation in order to drive further innovation. Examples of future research directions include the use of hiPSC-CMs to improve the modeling of sexually dimorphic disease mechanisms or scRNAseq to elucidate transcriptional differences between the sexes in response to pathology. A greater understanding of these mechanisms may enable the development of therapeutic or diagnostic tools to inform the clinical approach towards CVD across the sexes.

## Figures and Tables

**Figure 1 ijms-25-05964-f001:**
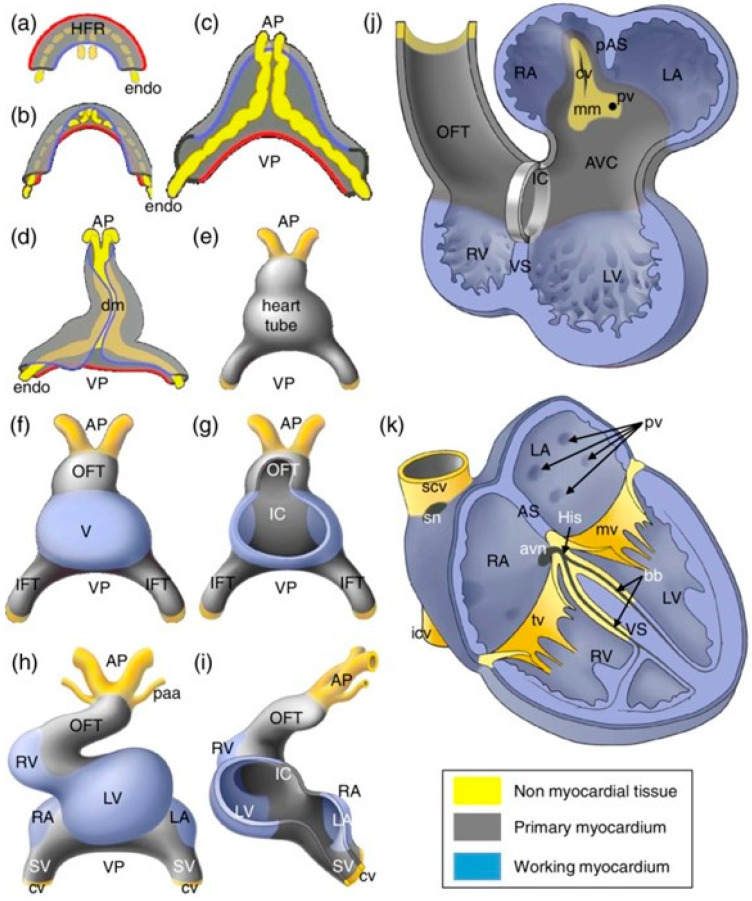
A schematic representation of the developmental stages of the human heart. Figure adapted from reference [14] with permission [14]. Blue: working myocardium, grey: primary myocardium, yellow: non myocardial tissue. (**a**–**k**): yellow lines indicate endocardial cells, blue lines indicate the medial border and red lines indicate the lateral border of the heart-forming region. Important abbreviations: HFR—heart-forming region, AP—arterial pole, VP—venous pole, dm—dorsal mesocardium, OFT—outflow tract, IFT—inflow tract, LA—left atrium, RA—right atrium, LV—left ventricle, RV—right ventricle, VS—ventricular septum, mv—mitral valve, and tv—tricuspid valve.

**Figure 2 ijms-25-05964-f002:**
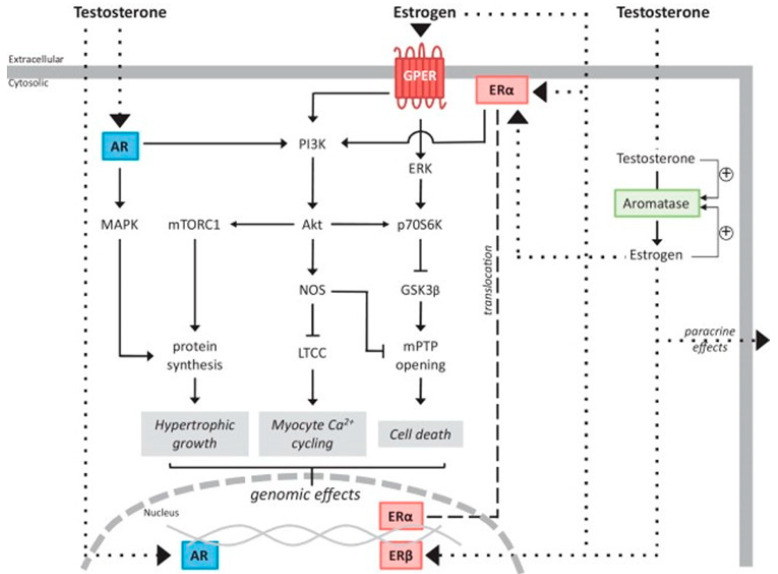
An outline of sex hormone pathways involving membrane-bound and nuclear receptors. Figure adapted from reference with permission [33]. This figure also includes sex hormone pathways that are not fully established in the heart. Dotted line: movement of the hormone. Dashed line: translocation of Erα to the nucleus.

**Figure 3 ijms-25-05964-f003:**
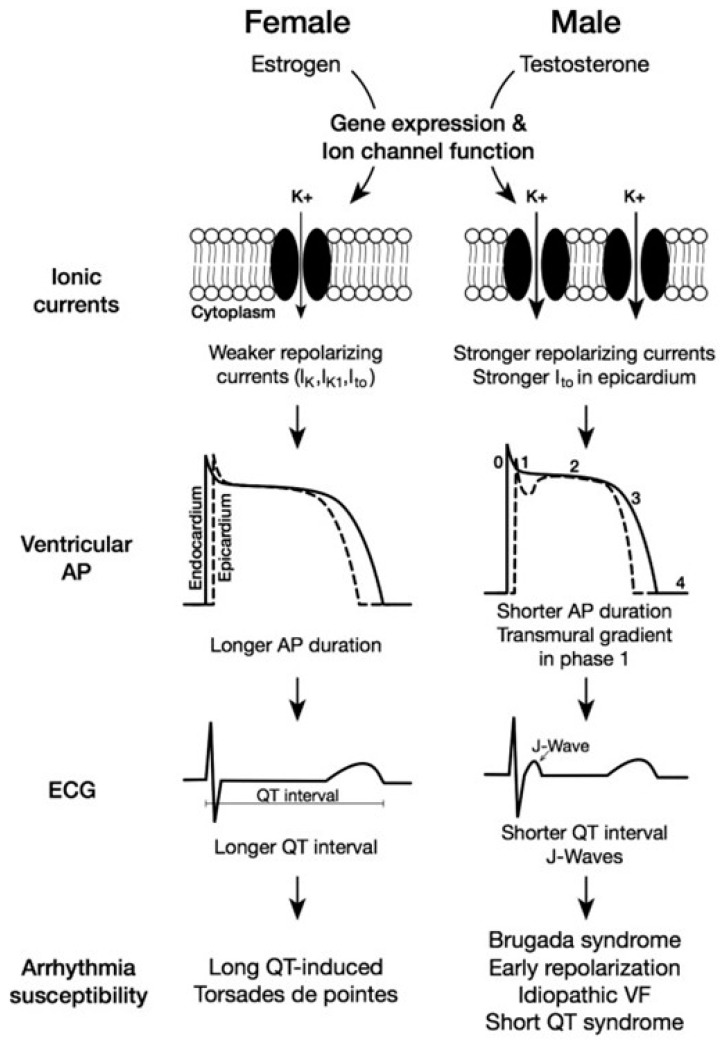
Mechanisms of sex-based differences in cardiac electrophysiology. Figure adapted from reference [99] with permission [101]. Both estrogen and testosterone regulate the expression of ion channels in the heart. Estrogen results in weaker repolarizing currents and longer QT interval in females, leading to higher risk for TdP. Testosterone strengthens repolarizing currents and results in shorter QT interval in males, which increases the risk for Brugada syndrome, early repolarization, idiopathic ventricular fibrillation, and short QT syndrome.

**Figure 4 ijms-25-05964-f004:**
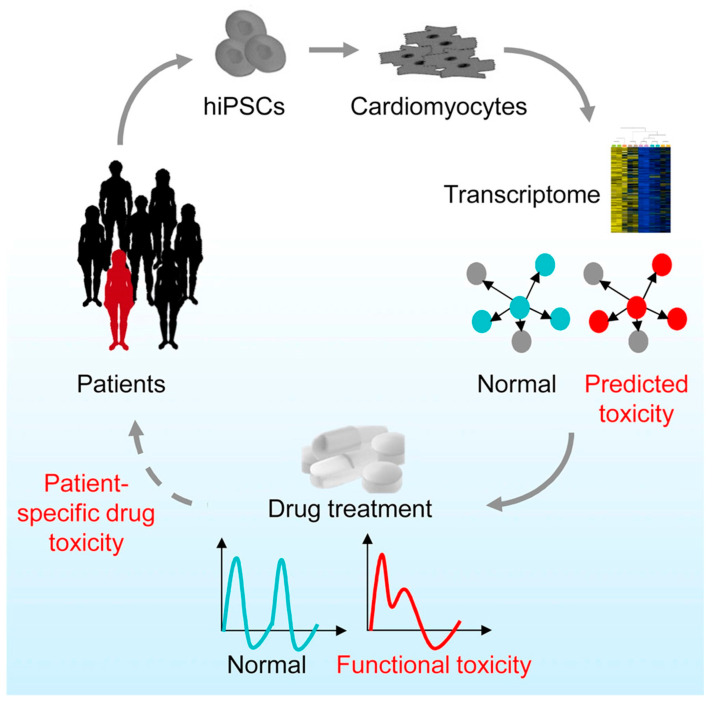
The use of hiPSC-CMs in predicting drug safety and efficacy in individual patients. Figure adapted from reference [110] with permission [112]. Patient-derived iPSC-CMs are used to test drug-induced toxicity and could serve as a personalized drug screening platform.

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
