# Peer review of "Sex-Based Mechanisms of Cardiac Development and Function: Applications for Induced-Pluripotent Stem Cell Derived-Cardiomyocytes"

_ijms, 2024, doi:10.3390/ijms25115964_

Round 1

Reviewer 1 Report

Comments and Suggestions for Authors

This is a very interesting review summarizing the main scientific articles dealing with sex-based mechanisms of cardiac development and functions, with a focus on sex hormones and their influence, and also the usefulness of iPSCs to study them.

The article is worth being published, but there are some minor issues, related mainly to referencing, but also there are some instances in which the information given to the reader should be more specific. 

Some examples are given below: 

- line 86 - which expressed cardiac proteins are specific to women only? Once introduced an idea, it should be followed.

- line 88 - which proteins are conserved in adulthood?

- lines 81-109 - reference 2 describes a study done using animal models. This should be stated.

- lines 110-119 - there should be references at least at the end of the paragraph

- line 120 - which studies? There should be references

- line 183 - there should be a reference

- line 199 - there should be a reference

- Figure 2 should have a higher resolution

Also, some references were used extensively (see. e.g. ref. 2). While this is not an issue per se, for a comprehensive review such as this, I would have liked to see tome more variability (e.g. going further on some ideas sampled from that references, and see what other similar studies were published on a similar topic).

Author Response

Thank you very much for taking the time to review this manuscript. Please find the detailed responses below and the corresponding revisions highlighted in the re-submitted files.

"The article is worth being published, but there are some minor issues, related mainly to referencing" and "I would have liked to see tome more variability (e.g. going further on some ideas sampled from that references, and see what other similar studies were published on a similar topic)." Thank you for pointing this out. To improve referencing, we did a literature search on the most recent publications on sex hormone and cardiac development and function. We found two papers that are very relevant to our topic and incorporated them into the manuscript (lines 88, 97, 112, 129, 452). The two references are "Regitz-Zagrosek et al. Nature Reviews Cardiology. 2023." and "McClain et al. Science Advances. 2024." In addition we added the missing references that were pointed out at the beginning and end of certain paragraphs (lines 131, 195, 211).

"but also there are some instances in which the information given to the reader should be more specific." We agree that some information presented are not specific. As recommended, we stated that the experiment in reference 2 was done in mouse models (line 81). We also included examples of differentially expressed proteins that are on the X chromosome and proteins that were conserved in adulthood (lines 92-97). In the last paragraph on page 10, we explained how that the DNA methylation data came from EWAS studies and DNA was extracted from peripheral blood to avoid confusion (lines 395-398). 

"Figure 2 should have a higher resolution." Thank you for pointing this out. However, this figure is taken and cited from "Bell et al. The Journal of Steroid Biochemistry and Molecular Biology. 2013." and it was the highest resolution we could download.

Reviewer 2 Report

Comments and Suggestions for Authors

The sex-based development, function and diseases of the heart is reviewed.  This is a nicely written review that presents various related topics in a logical manner, making it easy to read and follow.  I have two suggestions that may improve this article and one question.  1).  Figure legends are short and not really helpful for appreciating the figures.   For example, abbreviations are not defined, there are dotted and solid lines, and there are differently colored objects.  Authors should consider providing more explanations for each figure such as what different parts of each figure illustrates.  2)  It appears that the most recent papers are not cited.  For example, there is only one citation published in 2023 while there are many of those published in 2022.  The reference list could be updated.  In addition, two important recent reviews are not cited.  Please cite the following two reviews.  i) Regitz-Zagrosek V and Gebhard C. 2023. Gender medicine: effects of sex and gender on cardiovascular disease manifestation and outcomes. Nature Reviews Cardiology | Volume 20 | April 2023 | 236–247.  ii) Mcclain et al.  Sex in cardiovascular disease: Why this biological variable should be considered in in vitro models.  Science Advances  2024.  These reviews discuss topics that are directly related to this review.  Lines 379-380.  You write, “Testosterone introduction has been shown to cause changes in DNA methylation in the blood.”  This expression is a bit too loose.  Ordinarily, there is little DNA floating in the blood.  Do you mean cells in the blood?  At any rate, please explain more fully.

Author Response

Thank you very much for taking the time to review this manuscript. Please find the detailed responses below and the corresponding revisions highlighted in the re-submitted file.  

"Figure legends are short and not really helpful for appreciating the figures." We totally agree with your suggestion and updated all figure legends to explain the different colours, lines and abbreviations to make the information presented to readers easier to understand.   

"It appears that the most recent papers are not cited.  For example, there is only one citation published in 2023 while there are many of those published in 2022." Thank you for pointing this out. We incorporated the two papers that you suggested "Regitz-Zagrosek et al. Nature Reviews Cardiology. 2023." and "McClain et al. Science Advances. 2024." into the manuscript (lines 88, 97, 112, 129, 452). Both of these papers are very helpful in supporting ideas brought up in our manuscript. Since the manuscript only discusses sex as a biological variable, we focused on sex but not gender related information from Regitz-Zagrosek's paper. We also did a literature search of the recent publications on sex hormone and cardiac development and function. However, most of the papers focus on the effects of hormones on vascular function and are not as relevant to our manuscript.

"You write, “Testosterone introduction has been shown to cause changes in DNA methylation in the blood.”  This expression is a bit too loose.  Ordinarily, there is little DNA floating in the blood.  Do you mean cells in the blood?  At any rate, please explain more fully." We agree. To explain the concept clearly, we specified that DNA is extracted from peripheral blood and methylation data comes from EWAS.